# L-Arginine and Taurisolo^®^ Effects on Brain Hypoperfusion–Reperfusion Damage in Hypertensive Rats

**DOI:** 10.3390/ijms251910868

**Published:** 2024-10-09

**Authors:** Dominga Lapi, Gian Carlo Tenore, Giuseppe Federighi, Martina Chiurazzi, Santo Nunziato, Maria S. Lonardo, Mariano Stornaiuolo, Antonio Colantuoni, Ettore Novellino, Rossana Scuri

**Affiliations:** 1Department of Biology, University of Pisa, 56127 Pisa, Italy; dominga.lapi@unipi.it; 2Department of Pharmacy, University of Naples “Federico II”, 80138 Naples, Italy; giancarlo.tenore@unina.it (G.C.T.); mariano.stornaiuolo@unina.it (M.S.); 3Department of Translational Medicine and New Technologies in Medicine and Surgery, University of Pisa, 56127 Pisa, Italy; giuseppe.federighi@unipi.it; 4Department of Clinical Medicine and Surgery, University of Naples “Federico II”, 80138 Naples, Italy; martina.chiurazzi@unina.it (M.C.); santo.nunziato@gmail.com (S.N.); mslonardo91@gmail.com (M.S.L.); antonio.colantuoni@unina.it (A.C.); 5Hospital “A. Gemelli”, Catholic University, 00136 Rome, Italy; ettore.novellino@unicatt.it

**Keywords:** brain hypoperfusion–reperfusion injury, L-arginine, Taurisolo^®^, spontaneously hypertensive rat (SHR), dexamethasone-induced hypertensive rat (DIHR), pial microcirculation

## Abstract

Acute and chronic hypertension causes cerebral vasculopathy, increasing the risk of ischemia and stroke. Our study aimed to compare the effects of arterial pressure reduction on the pial microvascular responses induced by hypoperfusion and reperfusion in spontaneously hypertensive Wistar rats, desamethasone-induced hypertensive Wistar rats and age-matched normotensive Wistar rats fed for 3 months with a normal diet or normal diet supplemented with L-arginine or Taurisolo^®^ or L-arginine plus Taurisolo^®^. At the end of treatments, the rats were submitted to bilateral occlusion of common carotid arteries for 30 min and reperfusion. The microvascular parameters investigated in vivo through a cranial window were: arteriolar diameter changes, permeability increase, leukocyte adhesion to venular walls and percentage of capillaries perfused. Hypoperfusion–reperfusion caused in all rats marked microvascular changes. L-arginine treatment was effective in reducing arterial blood pressure causing vasodilation but did not significantly reduce the damage induced by hypoperfusion–reperfusion. Taurisolo^®^ treatment was less effective in reducing blood pressure but prevented microvascular damage from hypoperfusion–reperfusion. L-arginine plus Taurisolo^®^ maintained blood pressure levels within the physiological range and protected the pial microcirculation from hypoperfusion–reperfusion-induced microvascular injuries. Therefore, the blood pressure reduction is not the only fundamental aspect to protect the cerebral circulation from hypoperfusion–reperfusion damage.

## 1. Introduction

Arterial hypertension is a chronic disease affecting more than 1 billion people worldwide. The complications of the disease include cerebro-vascular stroke, coronary artery disease, heart failure and kidney diseases [1,2,3]. These are major causes of morbidity and mortality all around the world. The guidelines of the European Society of Cardiology and the European Society of Hypertension recommend investigating coexisting cardiovascular risk factors and sub-clinical target organ damage. However, arterial blood pressure reduction in hypertensive people can prevent or drastically reduce complications [4,5,6,7,8]. Recently, it was shown that reactive oxygen species (ROS) play an important role in the regulation of vascular function and in the development of cardiovascular and brain diseases [9,10]. Pathophysiologic processes in hypertension are linked to the risk factor oxidative stress and it is known that low concentrations of ROS are important for redox regulation in maintaining endothelium integrity and vascular function. In particular, a laminar flow increases endothelial nitric oxide synthase (eNOS) expression, activity and NO production, while during hypertension, oscillatory flow increases ROS formation and subsequent oxidative damage [11,12,13]. The type and the degree of shear stress can differentially regulate the ROS- and NO-dependent signaling and remodeling of the vasculature [14,15,16].

According to these references, the aim of the present study was to evaluate whether a reduction in the arterial blood pressure is a sufficient condition to prevent and/or reduce damage from ischemia and reperfusion or, according to the most recent studies, it is better to reduce the formation of ROS.

In the study, we evaluated the microvascular responses of the pial network in the parietal area to damage induced by bilateral occlusion of common carotid arteries for 30 min and reperfusion for 60 min in the rat model of brain hypoperfusion and reperfusion, using spontaneously hypertensive rats (SHRs), dexamethasone-induced hypertensive Wistar rats (DIHRs) and age-matched normotensive Wistar rats (NWRs) fed with a normal diet (control) or with normal diet supplemented with L-arginine or Taurisolo^®^ or L-arginine plus Taurisolo^®^.

Arterial blood pressure reduction can be due to NO whose synthesis is promoted by eNOS which uses L-arginine as a substrate [17,18,19,20]. Taurisolo^®^ is a nutraceutical obtained from the Aglianico grape which has antioxidant properties that reduce oxidative stress and preserve vascular function against the aging process [21,22]. As tested in humans [22], Taurisolo^®^ significantly reduces the concentration of trimethylamine-N-oxide (TMAO) in the blood. TMAO is an amine oxide derived from choline and L-carnitine that is metabolized by the gut microbiota, and it appears to be a highly oxidant and reactive molecule [23]. Although mechanistic studies have not been performed, according to the available literature, it is possible to hypothesize two mechanisms of action by which Taurisolo^®^ may exert its TMAO-reducing effect: antioxidant activity and microbiota remodeling, both exerted by polyphenols [24]. In the rat, we showed that Taurisolo^®^ induces pial arteriolar dilation and prevents ROS formation and significantly reduces infarct size. These effects were accompanied by an increase in eNOS expression. Moreover, mass-spectrometry metabolomics analysis detected a marked decrease in the amount of peroxidized cardiolipin and pronounced reduction in pro-inflammatory prostaglandins and thromboxane Txb2 [25].

Therefore, in this study, L-arginine and Taurisolo^®^ were also administered in association to investigate the interplay of the two different substances in the complex scenario of brain microvascular injury.

## 2. Results

### 2.1. Mean Arterial Blood Pressure (MABP)

The administration of L-arginine or Taurisolo^®^ or L-arginine plus Taurisolo^®^ to both SHRs and DIHRs induced a statistically significant decrease in MABP according to the treatment while in NWRs only the L-arginine treatment significantly reduced MABP. As shown in Table 1, the treatment with Taurisolo^®^ alone was less effective in reducing MABP in all groups of rats considered.

### 2.2. Arteriolar Diameter Changes

Previous studies [16] carried out in NWRs demonstrated that a 30-min hypoperfusion caused a significant reduction in pial arteriolar diameter. During the reperfusion, arterioles underwent a rapid succession of events: vasoconstriction, vasodilation and finally vasoconstriction that lasted at least for 60 min. In the present study, we analyzed the arteriolar diameter at 60 min of reperfusion. As expected, the control sub-group in each experimental group showed a significant diameter decrease (*p* < 0.0001) with respect to the baseline condition (basal, Figure 1), while all treatments caused a significant vasodilation in all groups (Figure 1) (*p* < 0.0001 vs. basal for all sub-groups). The diameter increase was greater in SHRs (Figure 1A) and DIHRs (Figure 1B) than in NWRs (Figure 1C) after L-arginine and L-arginine plus Taurisolo^®^ treatments. In the former case, in SHRs, there was a marked increase in arteriolar diameter by 38.0 ± 3.6% of baseline (*p* < 0.01 vs. basal), in DIHRs by 40.0 ± 3.8% of baseline (*p* < 001 vs. basal), while in NWRs the increase was 30.0 ± 2.5% of baseline (*p* < 0.01 vs. basal); in the latter case, in SHRs the arteriolar diameter increased by 30.1 ± 2.8% of baseline (*p* < 0.01 vs. basal), in DIHRs by 37.5 ± 2.6% of baseline, and in NWRs only by 15.0 ± 2.0% of baseline. The treatment with Taurisolo^®^ alone increased the arteriolar diameter less effectively even if statistically significant in SHRs (8.0 ± 1.2% of baseline, *p* < 0.05 vs. basal) and DIHRs (15.5 ± 0.4% of baseline, *p* < 0.01 vs. basal) than in NWRs (18.7 ± 2.5% of baseline, *p* < 0.0001 vs. basal) (Figure 1C).

The control rats showed a significant reduction in pial arterioles diameter compared with baseline at 60 min of reperfusion while the L-arginine (gray columns), Taurisolo^®^ (light gray columns) and L-arginine plus Taurisolo^®^ (white columns) treatments caused a significant increase in pial arterioles diameter compared with baseline.

### 2.3. Microvascular Permeability Changes

In basal conditions, both control SHRs and control DIHRs already showed a reduced integrity of pial microcirculation with respect to control NWRs (NGL: in SHRs 0.08 ± 0.02, *p* < 0.0001 vs. NWRs; in DIHRs 0.09 ± 0.00, *p* < 0.0001 vs. NWRs; and in NWRs 0.02 ± 0.00) (Figure 2). L-arginine treatment did not modify the permeability in SHRs (NGL: 0.07 ± 0.01, *p* = 0.9742 vs. control) (Figure 3A) and significantly increased it in DIHRs (NGL: 0.15 ± 0.01, *p* = 0.0002 vs. control) (Figure 2B) and NWRs (NGL: 0.10 ± 0.01) (Figure 2C). Taurisolo^®^ treatment alone and in association with L-arginine significantly decreased the microvascular permeability in both SHRs (NGL: 0.03 ± 0.00 and 0.02 ± 0.00, respectively; *p* < 0.0001 vs. control in both cases, Figure 2A) and DIHRs (NGL: 0.03 ± 0.00 and 0.05 ± 0.00, respectively; *p* < 0.001 vs. control in both cases, Figure 2B), while in NWRs the fluorescent leakage was reduced only with respect to L-arginine-treated rats (*p* < 0.009 and *p* < 0.0001, respectively) (Figure 2C).

At 60 min’ reperfusion, a marked increase in fluorescent leakage occurred in all the groups of rats considered. Interestingly, in control SHRs (NGL: 0.50 ± 0.03, *p* < 0.0001 vs. basal) and L-arginine-treated SHRs (NGL: 0.45 ± 0.01, *p* < 0.000 vs. basal), as well as in control DIHRs (NGL: 0.48 ± 0.01, *p* < 0.0001 vs. basal) and L-arginine-treated DIHRs (NGL: 0.49 ± 0.01, *p* < 0.0001 vs. basal), the increase was greater than in NWRs (NGL: 0.28 ± 0.01, control, *p* < 0.05 vs. SHRs and DIHRs; NGL: 0.38 ± 0.01, L-arginine-treated NWRs, *p* < 0.0001 vs. L-arginine-treated SHRs and L-arginine-treated DIHRs) (Figure 2), while Taurisolo^®^ treatment alone determined a slight increase in permeability and in association with L-arginine a moderate increase in fluorescent leakage both in SHRs (NGL: 0.10 ± 0.01, Taurisolo^®^-treated SHRs, *p* = 0.0004 vs. control; NGL: 0.13 ± 0.01, L-arginine plus Taurisolo^®^-treated SHRs, *p* = 0.0008 vs. control) and in DIHRs (NGL: 0.07 ± 0.01, Taurisolo^®^-treated DIHRs, *p* < 0.0001 vs. control; NGL: 0.16 ± 0.01, L-arginine plus Taurisolo^®^ -treated DIHRs, *p* < 0.0001 vs. control) with respect to the increase observed in NWRs (NGL: 0.13 ± 0.01, Taurisolo^®^-treated NWRs, *p* < 0.0001 vs. control; NGL: 0.28 ± 0.01, L-arginine plus Taurisolo^®^-treated NWRs, *p* = 0.999 vs. control).

### 2.4. Leukocyte Adhesion

In basal conditions, in control SHRs there was a significantly higher number of leukocytes adherent to venular walls ((13.00 ± 1.00/100 µm venular wall), *p* < 0.0001 vs. NWRs) (Figure 3A) than in NWRs (2.00 ± 0.30/100 µm venular wall). L-arginine treatment did not modify the leukocytes’ adhesion with respect to control in all three groups of rats considered, while the treatment with Taurisolo^®^ alone and in association with L-arginine produced a significant reduction of leukocytes adherent to venular walls only in SHRs ((5.00 ± 0.60/100 µm venular wall), in Taurisolo^®^-treated SHRs, *p* < 0.0001 vs. control; (6.00 ± 0.60/100 µm venular wall) in L-arginine plus Taurisolo^®^-treated SHRs, *p* = 0.0023 vs. control) (Figure 3A). At 60 min of reperfusion, the number of leukocytes adherent to venular walls significantly increased with respect to basal conditions in all groups of rats (Figure 3). L-arginine treatment significantly decreased the number of leukocytes adherent in both SHRs ((10.00 ± 0.80/100 µm venular wall), *p* = 0.023 vs. control) and DIHRs ((11.00 ± 0.40/100 µm venular wall), *p* = 0.0146 vs. control), while in NWRs it caused a significant augmentation ((11.00 ± 1.20/100 µm venular wall), *p* = 0.0254 vs. control). Taurisolo^®^ treatment alone ((5.00 ± 0.60/100 µm venular wall) in SHRs, *p* = 0.0023 vs. control; (3.00 ± 0.30/100 µm venular wall) in DIHRs, *p* < 0.0001 vs. control); ((3.00 ± 0.01/100 µm venular wall) in NWRs, *p* < 0.0001 vs. control) and in association with L-arginine ((6.00 ± 0.60/100 µm venular wall) in SHRs, *p* < 0.0001 vs. control; (8.00 ± 0.40/100 µm venular wall) in DIHRs, *p* < 0001 vs. control; (2.00 ± 0.30/100 µm venular wall) in NWRs, *p* < 0.0001 vs. control) determined a significant reduction in the leukocyte adhesion in all groups of rats considered with respect to controls (Figure 3).

### 2.5. Capillary Perfusion Changes

As previously described [25], at the end of reperfusion the capillary network was markedly impaired. A reduction in the number of capillaries perfused is evident in Figure 4 where in all control sub-groups the percentage of perfused capillaries was significantly reduced compared with the basal condition (*p* < 0.0001). The L-arginine treatment significantly improved the capillary perfusion only in NWRs (37.50 ± 1.50% reduction in control with respect to baseline, 32.00 ± 1.00% reduction in L-arginine-treated NWRs with respect to baseline, *p* < 0.0001) (Figure 4C), while the treatment with Taurisolo^®^ alone and in association with L-arginine preserved, although not totally, the capillary perfusion in all group of animals and particularly in SHRs (Figure 4A) and DIHRs (Figure 4B) (*p* < 0.0001). In particular, in Taurisolo^®^-treated SHRs, the reduction in perfused capillaries was by 5.50 ± 1.00% of baseline and in L-arginine and Taurisolo^®^-treated SHRs by 9.30 ± 1.50% of baseline. In Taurisolo^®^-treated DIHRs, the decrease in perfused capillaries was by 9.00 ± 1.00% of the baseline and in L-arginine plus Taurisolo^®^-treated DIHRs by 6.50 ± 1.00% of baseline.

At 60 min of reperfusion, the control rats (black columns) showed a significant reduction in perfused capillaries with respect to basal conditions. L-arginine treatment (gray columns) only slightly improved the quantity of perfused capillaries in DIHRs and NWRs compared with control. Taurisolo^®^ treatment (light gray columns) caused a significant increase in the capillaries perfused in all experimental groups with respect to control as well as L-arginine plus Taurisolo^®^ (white columns) treatment. No treatment results in complete recovery of the number of perfused capillaries.

## 3. Discussion

Brain hypoperfusion due to bilateral occlusion of common carotid arteries and the subsequent reperfusion are known to induce microvascular damage in NWRs [26]. The present data indicate that in SHRs and in a model of experimental hypertension, DIHRs, the hypoperfusion and subsequent reperfusion caused more marked alterations of pial vessel diameter changes, microvascular permeability, adhesion of leukocytes to venular walls and capillary perfusion. There were no significant differences in damage between SHRs and DIHRs and in particular in the crucial parameter, the reduction in the number of perfused capillaries. Comparing the data obtained in NWRs submitted to brain hypoperfusion–reperfusion with those in hypertensive rats, we can note that in both SHRs and DIHRs, there was a higher arteriolar diameter reduction than in NWRs, and the adhesion of leukocytes to the venular walls was enhanced in hypertensive animals. Our data support previous observations in SHRs and NWRs submitted to occlusion of the right middle cerebral artery (MCAO, 90 min) and reperfused for 24 h. Structural, mechanical and myogenic properties of the MCA were investigated by pressure myography. The results showed that in SHRs there was hypertrophic remodeling of the MCA, with a decrease in cross-sectional area during ischemia–reperfusion without changes in distensibility. On the other hand, in Wistar rats the MCA presented enhanced distensibility and decreased myogenic tone, facilitating cerebral perfusion. Furthermore, previous data indicate that in SHRs submitted to temporary MCAO and reperfusion there was marked neutrophil infiltration, contributing to vascular damage [27].

Our data show an increase in the microvascular permeability measured as leakage of fluorescent dextran in both SHRs and DIHRs with respect to NWRs. According to our results, some studies demonstrated that in both acute and chronic hypertension, permeability of the endothelium is enhanced. Chronic hypertension causes a hypertrophy of the endothelium characterized by an increase in cell volume, a distortion of the nuclear regions that protrude into the lumen of the vessel, an increase in tight junctions and an anomalous morphology of the gap junctions. The basal elastic lamina increased consequentially to an increase in the synthesis of connective tissue. The acute hypertension, indeed, causes endothelial cell proliferation which is very pronounced in the early phases of experimental hypertension [28].

The L-arginine treatment reduced the mean arterial blood pressure by inducing a marked vasodilation (Figure 1). L-arginine is the substrate of coupled eNOS and promotes NO production, ultimately reducing arterial blood pressure. As regards the damage induced by hypoperfusion and consequent reperfusion, hypertensive rats showed different responses with respect to normotensive rats to the L-arginine administration. After L-arginine treatment, in NWRs we observed an increase in the microvascular permeability (Figure 2), in the number of adherent leukocytes (Figure 3) and in the number of perfused capillaries (Figure 4). In SHRs and DIHRs L-arginine administration did not affect the capillaries’ perfusion, seems to prevent the increase in microvascular permeability only in SHRs, even if the data obtained are not statistically significant, and preserves the integrity of the vascular walls in all hypertensive rats, as the leukocytes’ adhesion was significantly reduced with respect to controls. These results can be explained by considering that eNOS presents two functional states, referred to as coupled and uncoupled, which enzymatically compete [29]. Uncoupled eNOS is implicated in the pathogenesis of vascular diseases, contributing to oxidative stress, inflammation, leukocytes’ adhesion and endothelial dysfunction, conditions indicative of the hypertensive state [30]. eNOS uncoupling can occur due to loss of L-arginine availability, oxidative stress, deficiency or altered redox states of eNOS cofactors. On the other hand, coupled eNOS is associated with cardiovascular protection, as NO produced in this state promotes vasodilation [19]. Our results indicate that L-arginine treatment caused a greater vasodilation in hypertensive than in normotensive rats (see # in Figure 1). Therefore, we can hypothesize that in our experimental models of hypertension, the activity of uncoupled eNOS prevails over the action of coupled eNOS and the administration of L-arginine induces a potentiation of coupled eNOS which determines beneficial microvascular effects. Further experiments will shed light on this hypothesis.

Interestingly, the treatment with Taurisolo^®^ alone was less efficient in reducing MABP than L-arginine treatment, but alone or in association with L-arginine it was very effective in protecting the pial microcirculation from the damage produced by hypoperfusion and subsequent reperfusion in both normotensive and hypertensive rats.

The main effects of Taurisolo^®^ are to keep arterioles dilated and reduce the formation of free radicals during the reperfusion period in order to ensure adequate capillary flow and consequently sufficient blood supply to the cortical tissue. The ability to induce vasodilation at the level of pial arterioles is due to NO release, as shown in our previous work [25]. Using the same experimental model, it was observed that in animals treated with Taurisolo^®^ there was a significant increase in eNOS expression in both cortex and striatum. The inhibition of eNOS with N5-(1-Iminoethyl)-L-ornithine dihydrochloride (L-NIO), a potent, irreversible inhibitor of eNOS endothelial nitric oxide synthase, blunted arteriolar dilation.

Moreover, all recent studies regarding the effects of Taurisolo^®^ agree that the substance has a powerful antioxidant effect [21,24,25,31]. In accordance with the literature, our preliminary studies carried out in NWRs indicate an important reduction of reactive oxygen species detected in our preparations by the administration of Taurisolo^®^ as shown in Figure 5. It is therefore possible to hypothesize that the administration of Taurisolo^®^ alone or in association with L-arginine might counteract or even prevail over the effects induced by uncoupled eNOS.

The results of the present study indicate that arterial blood hypertension can worsen microvascular hypoperfusion and reperfusion damage in hypertensive animal models. However, it is conceivable to suggest that protection from damage due to hypoperfusion–reperfusion can be improved not only by a reduction in arterial blood pressure but also by molecules, such as Taurisolo^®^, effective in preserving the integrity of the brain–blood barrier, facilitating cerebral blood flow and preventing the reduction of capillary perfusion, as well as the damage due to adhesion of leukocytes to venular walls. Interestingly, the parameters specifically ameliorated by Taurisolo^®^ (and not by L-arginine) seem to point toward the bioactive fraction of this nutraceutical acting mostly by inhibiting uncoupled eNOS. It has indeed been shown that antioxidants are able to inhibit uncoupled eNOS either by mitigating oxidative stress or by re-establishing the correct redox state of eNOS cofactors (tetrahydrobiopterin, BH4) or of its iron heme moiety, the latter coordinating oxygen molecules during the conversion of L-arginine to NO.

While this hypothesis requires further experimental evidence, the scenario where L-arginine and Taurisolo^®^ might act on coupled and uncoupled eNOS, respectively, could explain the synergism and the complementary effects of the two supplements we measured in our study.

## 4. Materials and Methods

### 4.1. Animals

The animals used were treated according to the Guide for the Care and Use of Laboratory Animals of the National Institute of Health. The protocol was approved by the Committee on the Ethics of Animal Experiments of Pisa University and Italian Health Ministry (Permit Number: 156/2017-PR).

The experiments were carried out on 48 animals randomly divided into 3 different groups: the first one was SHRs (n = 16), consisting of male spontaneously hypertensive rats, arriving at our lab at 5 weeks old, which after 3–4 weeks of housing started to be fed with different supplemented diets (see below). At this age, SHRs show a full-blown hypertensive state (MABP: 175.0 ± 1.4 mmHg). The second one was DIHRs (n = 16), consisting of age-matched Wistar rats, treated with dexamethasone (20 µg/kg/die) by subcutaneous administration for 7 days to induce experimental hypertension (MABP: 182.0 ± 1.6 mmHg) [32]. The third one was NWRs (n = 16), age-matched Wistar rats with MABP of 122.0 ± 0.7 mmHg. SHRs, DIHRs and NWRs were divided into 4 sub-groups: the first (n = 4 for each sub-group) was fed with an L-arginine-supplemented (10 mg/kg/die) diet for 3 months [33]; the second sub-group (n = 4 for each sub-group) was fed with a Taurisolo^®^-supplemented (20 mg/kg/die) for 3 months [25] diet; the third sub-group (n = 4 for each sub-group) was fed with an L-arginine (10 mg/kg/die) plus Taurisolo^®^ (20 mg/kg/die)-supplemented diet for 3 months and the fourth sub-group (control) (n = 4 for each sub-group) was fed with a normal diet.

At the end of treatment, the animals of all sub-groups were submitted to hypoperfusion by bilateral occlusion of common carotid arteries for 30 min, followed by 60 min of reperfusion by removing the arterial occlusion.

### 4.2. Surgical Animal Preparation

The animals, before undergoing the surgical procedure, were anesthetized with α-chloralose utilizing the dosage of 60 mg/kg b.w., i.p. for induction and the dosage of 20 mg/kg b.w., i.v., every hour, for maintenance [34]. Successively, the animals were subject to tracheostomy and mechanically ventilated with room air and supplemental oxygen afterward having been paralyzed with tubocurarine chloride (1 mg/kg·h, i.v.). The respirator was adjusted to maintain blood gases within physiological range. End-tidal CO_2_ was continuously monitored. Small blood samples (0.2 mL) were periodically taken through a catheter placed in the femoral artery for blood gas analysis. A catheter was inserted into the femoral vein to administer supplemental anesthetic doses and fluorescence tracers: fluorescein isothiocyanate bound to dextran, molecular weight 70 kDa (FD 70), 50 mg/100 g b.w., i.v. as 5% wt/vol solution in 3 min; FITC, to visualize the pial microvascular network and rhodamine 6 G, 1 mg/100 g b.w. in 0.3 mL (final volume 0.3 mL·100 g^−1^·h^−1^) to label leukocytes.

Hypoperfusion was induced through clamping of the common carotid arteries for 30 min, then clamping was removed and the pial microcirculation was observed for 60 min, the reperfusion period, at the end of which the microvascular alterations are evident.

The rats were positioned on a special heated stereotaxic support to maintain a constant body temperature (37.0 ± 0.5 °C), checked through a rectal probe. A closed cranial window of 4 mm × 5 mm was implanted above the left frontoparietal cortex (stereotactic coordinates: posterior 1.5 mm to bregma; lateral, 3 mm to the midline) [35]. During the drilling of the cerebral cortex a cold saline solution was suffused on the skull to avoid overheating. The skull and the underlying dura mater were removed, and the window was equipped with two needles secured with dental cement to ensure constant inflow and outflow of aCSF to the brain parenchyma [36,37]. The rate of superfusion was 0.5 mL/min controlled by a peristaltic pump. During superfusion the intracranial pressure (ICP) was maintained at 5 ± 1 mmHg and measured by a pressure transducer connected to a computer. The composition of the aCSF was: 119.0 mM NaCl, 2.5 mM KCl, 1.3 mM MgSO_4_·7H_2_O, 1.0 mM NaH_2_PO_4_, 26.2 mM NaHCO_3_, 2.5 mM CaCl_2_ and 11.0 mM glucose (equilibrated with 10.0% O_2_, 6.0% CO_2_ and 84.0% N_2_; pH 7.38 ± 0.02). The temperature was maintained at 37.0 ± 0.5 °C.

The pial microcirculation was in vivo observed by a fluorescence microscopic technique. The fluorescence microscope (Leitz Orthoplan, Rocklin, CA, USA) was equipped with long-distance objectives (2.5×, numerical aperture (NA) 0.08; 10×, NA 0.20; 20×, NA 0.25; 32×, NA 0.40), a 10× eyepiece and a filter block (Ploemopak, Leitz, Oberkochen, Germany). A mercury lamp (100 W) provides the epiillumination; moreover, the microscope provided appropriate filters for FITC, for rhodamine 6G and a heat filter (Leitz KG1). The pial microvasculature was visualized with a DAGE MTI 300RC low-light-level digital camera and recorded by a computer-based frame grabber (Pinnacle DC 10 plus, Avid Technology, Burlington, MA, USA).

The arterial blood pressure was measured in all animals by a tail-cuff method (IITC, Life Science Inc., Los Angeles, CA, USA) and the values of MABP were reported.

### 4.3. Quantification of Microvascular Parameters

In all experimental preparations we have measured diameter and length of arterioles and classified them in orders by Strahler’s method [38], utilizing a frame by frame computerized method (Microvascular Imaging Program, MIP).

In baseline conditions and at the end of reperfusion, the arteriolar diameter changes were evaluated. This parameter was quantified for all orders of arterioles but, in the present paper, we report only the variations measured in order 3 arterioles. These vessels are more numerous compared to those of the other orders and shown to be more responsive, and for them it is possible to observe their entire course.

The integrity of pial microcirculation was measured by estimating fluorescent dextran extravasation from venules and expressed as normalized gray levels (NGLs): NGL = (I − Ir)/Ir, where Ir was the baseline gray level at the end of microvasculature filling with fluorescence and I was the value at the end of reperfusion. The MIP obtains gray levels, averaging data derived from five windows, measuring 50 × 50 μm (10× objective) and located outside the venules. To identify the same regions of interest, a computer-assisted device was used for XY movement of the microscope table.

The number of leukocytes adherent to the vessel walls was counted under baseline conditions and at the end of reperfusion, and it was reported as the number of adherent leukocytes/100 μm of venular length (v.l.)/30 s, utilizing apposite magnification.

The perfusion of capillaries was evaluated as the percentage of the capillaries showing blood flow assessed by MIP in an area of 150 × 150 μm; by the fluorescence microscopic technique a capillary was patent when the tracer flows into the vessel, while a non-perfused capillary appears dark (non-fluorescent); therefore, in the same area it is possible to quantify the number of perfused and non-perfused capillaries.

Finally, in vivo ROS production was evaluated by 2′–7′ dichlorofluorescein diacetate (DCFH-A), 250 µM, dissolved in artificial cerebro-spinal fluid (aCSF) superfused on the pial layer at 37.0 ± 0.5 °C. DCFH-DA is a lipophilic substance and a stable non-fluorescent probe with a high cellular permeability. When DCFH-DA reacts with intracellular radicals it converts to its fluorescent product (DCF) [39]. The remaining extracellular DCFH-DA was washed out with aCSF and the intensity of DCF fluorescence is proportional to the intracellular ROS level. The use of a dedicated filter (522 nm) permitted evaluation of the fluorescence intensity. The test allows us to detect the oxidative species at the end of hypoperfusion and reperfusion (n = 3 for each experimental group), quantified in NGL.

All measurements were performed by two blind operators and were compared to avoid bias due to single-operator judgement. In all cases the results overlapped.

### 4.4. Drugs

Unless otherwise stated, all drugs were purchased from Sigma-Aldrich (St. Louis, MO, USA) while Taurisolo^®^ was obtained from MBMed (Turin, Italy).

### 4.5. Statistical Analysis

Data are expressed as mean ± standard error (SE). The Kolmogorov–Smirnov test was used to verify if the data were normally distributed. Two-way ANOVA for repeated measures was used to compare MABP, the diameter changes, the increase in microvascular permeability (NGL), the leukocytes’ adhesion and the capillary perfusion data within each group and among the different sub-groups, considering the time and treatment factors and the interaction “time by treatment”. Post hoc tests were performed to analyze the significant differences among the sub-groups at each time considered, using Tukey’s multiple comparisons test, and to analyze the significance between times within each sub-group using Sidak’s multiple comparisons test.

These analyses were carried out by using GraphPad Prism 7.0 software (Software Inc., San Diego, CA, USA). Significance was set at *p* < 0.05.

## Figures and Tables

**Figure 1 ijms-25-10868-f001:**
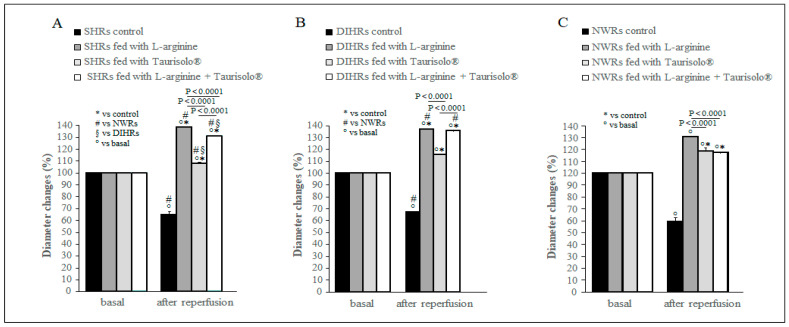
Modifications of pial arterioles diameter. (**A**) SHRs, (**B**) DIHRs and (**C**) NWRs. For each group the effects of the different treatments, normal diet (control), L-arginine, Taurisolo^®^ and L-arginine plus Taurisolo^®^, on vessel diameter at 60 min of reperfusion are shown. The data obtained in order 3 arterioles are plotted. The diameter changes are reported as percentage of baseline (basal) taken as 100%.

**Figure 2 ijms-25-10868-f002:**
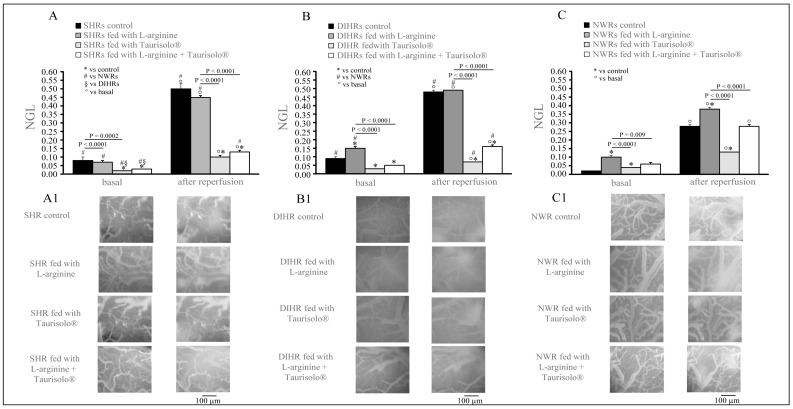
Microvascular permeability. (**A**) SHRs, (**B**) DIHRs, and (**C**) NWRs. The increase in microvascular permeability is reported as normalized gray levels (NGLs) in basal conditions (basal) and at 60 min of reperfusion. For each group, the effects of the different treatments, normal diet (control), L-arginine, Taurisolo^®^ and L -arginine plus Taurisolo^®^ are shown. Interestingly, in basal conditions and at 60 min of reperfusion, control SHRs and control DIHRs showed a significant increase in microvascular permeability compared with control NWRs (black column); L-arginine treatment (gray columns) did not significantly prevent the increase in leakage compared with control, while Taurisolo^®^ (light gray columns) and L-arginine plus Taurisolo^®^ (white columns) treatments protected pial microvasculature, reducing the microvascular permeability more with respect to the other treatments also in NWRs. (**A1**–**C1**) are computer-assisted images of pial microvascular networks in an SHR, DIHR and NWR, respectively, for each sub-group in basal conditions and at 60 min’ reperfusion.

**Figure 3 ijms-25-10868-f003:**
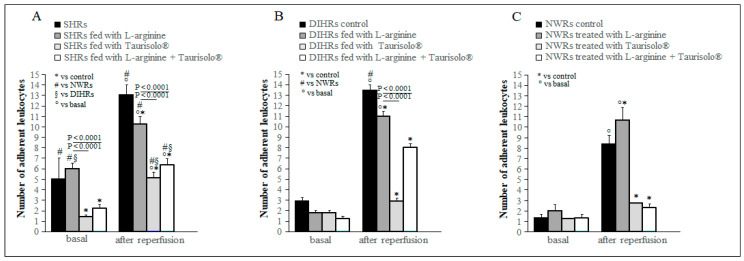
Leukocyte adhesion to venular walls. (**A**) SHRs, (**B**) DIHRs and (**C**) NWRs. The increase in leukocyte adhesion, expressed as number of adherent leukocytes/100 μm of venular length/30 s is plotted in basal conditions (basal) and at 60 min of reperfusion. For each group, the effects of the different treatments are shown. In basal conditions, control SHRs and control DIHRs showed an increased leukocyte adhesion compared with control NWRs, as well as at 60 min’ reperfusion. At 60 min’ reperfusion, L-arginine treatment (gray columns) induced a decrease in leukocyte adhesion in SHRs and DIHRs compared with control while an increase occurred in NWRs. Taurisolo^®^ (light gray columns) and L-arginine plus Taurisolo^®^ (white columns) treatments produced a reduction in the leukocyte adhesion with respect to L-arginine-treated rats (gray columns) and control rats (black column) in all experimental groups in both basal conditions and at 60 min’ reperfusion.

**Figure 4 ijms-25-10868-f004:**
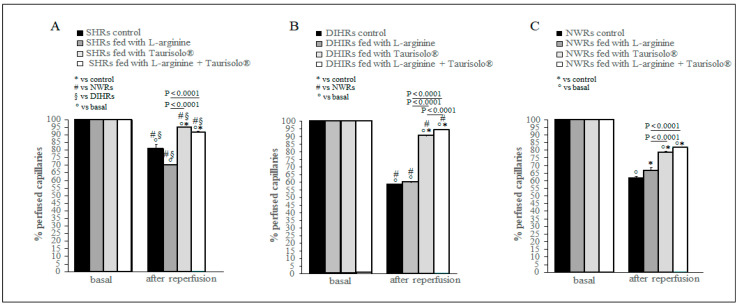
Perfused capillaries. (**A**) SHRs, (**B**) DIHRs and (**C**) NWRs. The number of perfused capillaries at 60 min of reperfusion is reported as percentage of the number of capillaries perfused in basal conditions taken as 100%. For each group the effects of all different treatments are shown.

**Figure 5 ijms-25-10868-f005:**
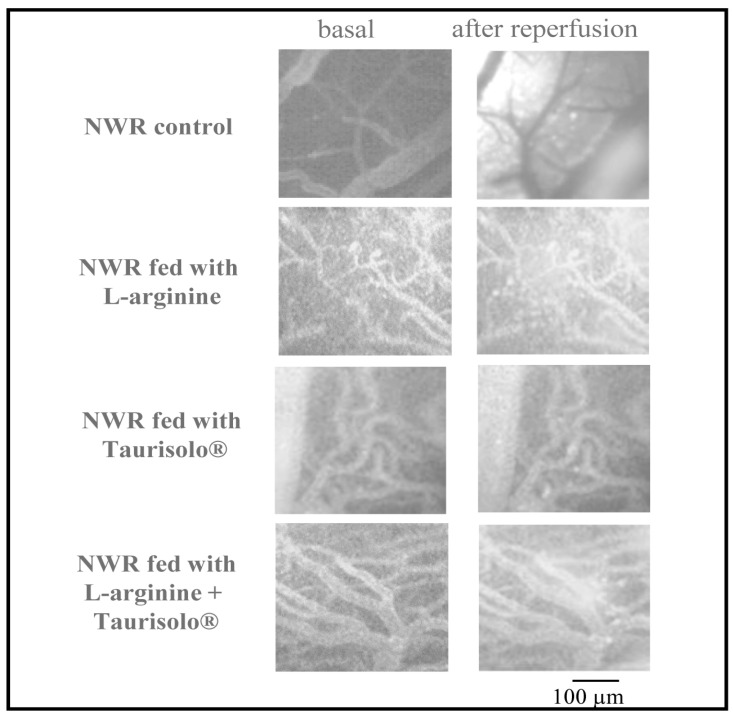
ROS detection. Computer-assisted images of pial microvascular networks in an NWR for each sub-group considered, captured in basal conditions and after 60 min’ reperfusion. Taurisolo^®^ treatment was more effective in preventing reactive oxygen species formation after 60 min’ reperfusion with respect to all the other treatments.

**Table 1 ijms-25-10868-t001:** MABP values recorded in the different rat sub-groups considered. * indicates significant difference between after and before treatment in each sub-group. ^⬪^ indicates significant difference between control and treated rats within each group.

Animal Sub-Groups	Mean Arterial Blood Pressure (MABP) mmHg	
	Before Treatment	After Treatment
SHRs (control)	175 ± 1.40	193 ± 4.00 *
SHRs FED WITH L-ARGININE	178 ± 2.00	130 ± 2.00 *^⬪^
SHRs FED WITH TAURISOLO	175 ± 3.90	160 ± 3.60 *^⬪^
SHRs FED WITH L-ARGININE + TAURISOLO	171 ± 2.00	132 ± 5.40 *^⬪^
DIHRs (control)	123 ± 1.03	182 ± 1.58 *
DIHRs FED WITH L-ARGININE	123 ± 1.04	125 ± 1.08 ^⬪^
DIHRs FED WITH TAURISOLO	123 ± 1.19	167 ± 1.03 *^⬪^
DIHRs FED WITH L-ARGININE + TAURISOLO	122 ± 0.75	121 ± 1.47 ^⬪^
NWRs (control)	122 ± 0.71	127 ± 4.02
NWRs FED WITH L-ARGININE	121 ± 0.63	110 ± 0.82 *^⬪^
NWRs FED WITH TAURISOLO	124 ± 1.03	122 ± 0.63
NWRs FED WITH L-ARGININE + TAURISOLO	124 ± 0.95	118 ± 1.03 ^⬪^

## Data Availability

Data are contained within the article.

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
