# Peer review of "L-Arginine and Taurisolo^®^ Effects on Brain Hypoperfusion–Reperfusion Damage in Hypertensive Rats"

_ijms, 2024, doi:10.3390/ijms251910868_

Round 1

Reviewer 1 Report

Comments and Suggestions for Authors

Reviewing the manuscript entitled, “L-Arginine and Taurisolo® Effects on Brain 2 Hypoperfusion-Reperfusion Damage in Hypertensive Rats” by Lapi D et al, this focuses on microvascular protection of Taurisolo® by the in vivo experiment model of several gene modified rats. Although this is an interesting manuscript, some of the information about Taurisolo® is too unscientific. At least in this manuscript, there is no direct evidence that this effect is due to Taurisolo®. The authors need to respond to the following concerns.

 First of all, the authors should describe what factors directly affect micro blood vessels when administered Taurisolo®? In addition, is there direct evidence that the results obtained with Taurisolo® are due to administration of Taurisolo®?

 The authors should explain your model, i.e. the microvascular protective effect of Taurisolo®, with accompanying figure.

 How was the effect of L-arginine (10 mg/kg/die) and Taurisolo® (20 mg/kg/die) determined?

 The authors mentioned “The diameter increase resulted greater in SHRs (Figure 1A) and DIHRs (Figure 1B) than in NWRs (Figure 1C) after L-arginine and L-arginine plus Taurisolo® treatments: in the former case, in SHRs, there was a marked increase of arteriolar diameter by 38.0 ± 3.6% of baseline (p < 0.01 vs. basal), in DIHRs by 40.0 ± 3.8% of baseline (p < 001 vs. basal) while in NWRs the increase was 30.0 ± 2.5% of baseline (p < 0.01 vs. basal); in the latter case, in SHRs the arteriolar diameter increased by 30.1 ± 2.8% of baseline (p < 0.01 vs. basal), in DIHRs by 37.5 ± 2.6% of baseline and in NWRs only by 15.0 ± 2.0% of baseline.” Is this statistically significant?

 In Figure 1, Taurisolo® significantly dilates arterioles, even though it does not have the effect of lowering mean blood pressure. What is the mechanism behind this?

 In Discussion, the authors mentioned “but prevented the increase of microvascular permeability only in SHRs and the leukocytes adhesion in all hypertensive rats.” From line 251 to 252. I cannot understand this at all. The authors should explain this in detail.

 In Discussion, from line 252 to 263 about eNOS, this statement is clearly jumping into conclusion. The authors should explain it using the results of this manuscript.

 The photos in Fig. 2 and Fig. 5 are extremely small and difficult to evaluate. The authors should modify them.

Author Response

Comments 1: First of all, the authors should describe what factors directly affect micro blood vessels when administered Taurisolo®? In addition, is there direct evidence that the results obtained with Taurisolo® are due to administration of Taurisolo®?

Response 1:

The effects of intravenous and oral Taurisolo® administration on microvasculature are described in details in a previous paper that we cited in the present work (reference 25).

In the present paper, within all groups of animals considered, we compare the effects of oral Taurisolo® administration with a control sub-group, consisting of untreated rats. Therefore, the observed effects can be ascribed to Taurisolo® treatment.

Comments 2: The authors should explain your model, i.e. the microvascular protective effect of Taurisolo®, with accompanying figure.

Response 2:

Images describing what is required are reported in reference 25.

Comments 3: How was the effect of L-arginine (10 mg/kg/die) and Taurisolo® (20 mg/kg/die) determined?

Response 3:

Taurisolo® dosage was determined by pilot experiments, as reported in our previous paper (reference 25, pag.3).  20 mg/Kg/die was the dosage which induced different effects evaluated with various experimental techniques, as reported in reference 25.

The effects of 10mg/kg/die of L-arginine are widely described in the literature, and we reported in the present paper reference 33.

Comments 4: The authors mentioned “The diameter increase resulted greater in SHRs (Figure 1A) and DIHRs (Figure 1B) than in NWRs (Figure 1C) after L-arginine and L-arginine plus Taurisolo® treatments: in the former case, in SHRs, there was a marked increase of arteriolar diameter by 38.0 ± 3.6% of baseline (p < 0.01 vs. basal), in DIHRs by 40.0 ± 3.8% of baseline (p < 001 vs. basal) while in NWRs the increase was 30.0 ± 2.5% of baseline (p < 0.01 vs. basal); in the latter case, in SHRs the arteriolar diameter increased by 30.1 ± 2.8% of baseline (p < 0.01 vs. basal), in DIHRs by 37.5 ± 2.6% of baseline and in NWRs only by 15.0 ± 2.0% of baseline.” Is this statistically significant?

Response 4:

Yes. As reported in Figure 1, # represents the statistical significance with respect to NWRs for both SHRs and DIHRs.

Comments 5:  In Figure 1, Taurisolo® significantly dilates arterioles, even though it does not have the effect of lowering mean blood pressure. What is the mechanism behind this?

Response 5:

Tank you for pointing this out. The answer lies in a better reading of Table 1. In the table we reported the MABP values recorded in the different rat sub-groups considered, and for all the animal groups we can compare in the “after treatment” column the value recorded after each treatment with the values recorded in the controls (animals that have not undergone the treatment). As can be seen, in each group of animals, the MABP value recorded after treatment with Taurisolo® is significantly lower than that recorded in controls. This means that Taurisolo® reduces arterial blood pressure although to a lesser extent than, for example, L-arginine. This effect can be ascribed to an increase in eNOS expression induced by Taurisolo® treatment, as we reported in our previous paper (reference 25).

Comments 6: In Discussion, the authors mentioned “but prevented the increase of microvascular permeability only in SHRs and the leukocytes adhesion in all hypertensive rats.” From line 251 to 252. I cannot understand this at all. The authors should explain this in detail.

Response 6:

We have accordingly revised this part of discussion, explaining in more detail our results. Page 8; lines: 260-264.

Comments 7: In Discussion, from line 252 to 263 about eNOS, this statement is clearly jumping into conclusion. The authors should explain it using the results of this manuscript.

Response 7:

In agreement with the suggestion, we modified the text referring to our results. Page 8; lines: 271 -273.

Comments 8: The photos in Fig. 2 and Fig. 5 are extremely small and difficult to evaluate. The authors should modify them.

Response 8:

To make Figure 2 and Figure 5 more readable, we enlarged the photos.

Reviewer 2 Report

Comments and Suggestions for Authors

I appreciate the authors for presenting this research article, which highlights the effects of L-arginine and Taurisolo® on hypoperfusion and reperfusion under different hypertensive conditions. My comments are as follows:

Introduction:

The introduction is not well-written and needs improvement.

(1) Some sentences are too long and contain too much information, which may confuse readers. For example: "Arterial blood pressure reduction is promoted, among others, by the enzyme endothelial nitric oxide synthase (eNOS) [6,7]. Using L-arginine..."

(2) The introduction goes into excessive detail about TMAO, gut microbiota, and Taurisolo®'s mechanisms, which could be simplified.

(3) The study’s hypothesis should be stated more clearly.

Methods:

In the dexamethasone group, the blood pressure is around 123 mmHg before treatment. What is the definition of hypertension in rats in terms of blood pressure?

Could the authors provide more information about the cranial window? What is its radius and exact location (coordinates)?

Results:

What is the mortality rate in each group?

Did the authors perform any neurological evaluations?

Discussion:

What are the limitations of the current study?

Author Response

Comments 1:

Introduction:

The introduction is not well-written and needs improvement.

(1) Some sentences are too long and contain too much information, which may confuse readers. For example: "Arterial blood pressure reduction is promoted, among others, by the enzyme endothelial nitric oxide synthase (eNOS) [6,7]. Using L-arginine..."

(2) The introduction goes into excessive detail about TMAO, gut microbiota, and Taurisolo®'s mechanisms, which could be simplified.

(3) The study’s hypothesis should be stated more clearly.

Response 1:

In agreement with the suggestions, we rewrote the introduction in a more orderly manner removing excessive details especially about the TMAO and indicating more clearly the study’s hypothesis. Introduction, page 1; lines 41 – 45; page 2; lines: 46 -53; page 2; lines: 66 -67 and page 2; lines:72 -77.

Comments 2:

Methods:

In the dexamethasone group, the blood pressure is around 123 mmHg before treatment. What is the definition of hypertension in rats in terms of blood pressure?

Could the authors provide more information about the cranial window? What is its radius and exact location (coordinates)?

Response 2:

In DIHR group, the hypertensive state was induced in the 7 days prior to the end of each treatment.Therefore, the MABPvalue reported in table 1,column Before treatment, corresponds to the value of arterial blood pressure in normotensive conditions.

Following the suggestion, we provided more detailed information about the cranial window. Results, page 10; lines: 350 -361.

Comments 3:

Results:

What is the mortality rate in each group?

Did the authors perform any neurological evaluations?

Response 3:

We didn't have any dead rats in any sub-groups.

At hte moment no neurological evaluation has been done.

Comments 4:

 Discussion:

What are the limitations of the current study?

Response 4:

As reported in page 8, line 276, our study needs further experimental investigations about the effects of antioxidants such as Taurisolo® on the activity of uncoupled and coupled eNOS in the acute and chronic hypertension.

Reviewer 3 Report

Comments and Suggestions for Authors

The work entitled “L-Arginine and Taurisolo Effects on BrainHypoperfusion-Reperfusion Damage in Hypertensive Rats” by Dominga Lapi et al. aimed to compare the effects of arterial pressure reduction on the pial microvascular responses induced by hypoperfusion and reperfusion feeding the rats for 3 months with normal diet, or normal diet supplemented with L-arginine or  Taurisolo or L-arginine plus Taurisol. 

The topic is very interesting and could make a valuable contribution in this regard, given the limited data in the literature. The work does not present any major criticalities.

Therefore, I believe it needs the following minor revisions, as outlined below:

- The authors should rewrite introduction in a more orderly manner. some concepts are confusing and unclear and are not adequately explored. it is advisable to revise the organization of the entire introduction, so as to also make clear the authors' motive for carrying out this study; in addition, increase the introduction on the main points of the work, dwelling on the factors considered later in the study;

- The authors should add the total number of rats used in their study and how they choose the number. It would be appropriate to determine it using a specific software (i.e. G*Power 3.1 software). Please clarify;

- There are no references referring to some of the experimental protocol used. Please add;

- Please provide higher-quality images; The specific area where computer-assisted images were taken is not specified. Please clarify.

- I recommend increasing the number of references, especially for the protocols in material and methods paragraph;

- About the results of figure 5, why do the authors analyze the production of microcirculatory ROS only in the NWR group and not also in all the others as in the previous analyses? In addition, in vivo ROS production was only evaluated in 3 animals for each experimental group); the authors should explain why they did not decide to use all 4 animals in each group Extend the conclusions, given the number of supporting results, while remaining within the allowed limit;

- Finally, review the English grammar and syntax for the entire text. In fact, there are incorrect English syntax periods as they are too long. Please carefully review the whole text along the article and correct it to make it easier to read. I suggest having the text reread and proofread by a native English speaker.

Comments on the Quality of English Language

- Finally, review the English grammar and syntax for the entire text. In fact, there are incorrect English syntax periods as they are too long. Please carefully review the whole text along the article and correct it to make it easier to read. I suggest having the text reread and proofread by a native English speaker.

Author Response

Comments 1: The authors should rewrite introduction in a more orderly manner. some concepts are confusing and unclear and are not adequately explored. it is advisable to revise the organization of the entire introduction, so as to also make clear the authors' motive for carrying out this study; in addition, increase the introduction on the main points of the work, dwelling on the factors considered later in the study.

Response 1:

In agreement with the suggestions, we rewrote the introduction in a more orderly manner removing excessive details especially about the TMAO and indicating more clearly the study’s hypothesis. Introduction, page 1; lines 41 –45; page 2; lines: 46 -53; page 2; lines: 66 -67 and page 2; lines:72 -77.

Comments 2: The authors should add the total number of rats used in their study and how they choose the number. It would be appropriate to determine it using a specific software (i.e. G*Power 3.1 software). Please clarify.

Response 2:

In agreement with the suggestion, we added the total number of rats used, page 10; lines: 316, 317, 320 and 323. This number of animals was authorized by the Committee on the Ethics of Animal Experiments of the Pisa University and Italian Health Ministry based on the indications provided by the software G*Power.

Comments 3: There are no references referring to some of the experimental protocol used. Please add.

Response 3:

Following the recommendation, we increased the number of references for the protocols in material and methods paragraph. References: 34, 35, 36, 37 and 39.

Comments 4: Please provide higher-quality images; The specific area where computer-assisted images were taken is not specified. Please clarify.

Response 4:

To make Figure 2 and Figure 5 more readable, we enlarged the photos. In material and methods paragraph, we specified the area where the images were taken. Page 10 ; lines: 353 -354.

Comments 5: I recommend increasing the number of references, especially for the protocols in material and methods paragraph.

Response 5:

See response to comment 3.

Comments 6: About the results of figure 5, why do the authors analyze the production of microcirculatory ROS only in the NWR group and not also in all the others as in the previous analyses? In addition, in vivo ROS production was only evaluated in 3 animals for each experimental group); the authors should explain why they did not decide to use all 4 animals in each group Extend the conclusions, given the number of supporting results, while remaining within the allowed limit.

Response 6:

In Figure 5 we reported the results of preliminary experiments performed in a group of animals other than the experimental ones to avoid interference between the various tracers used. A more exhaustive description of the antioxidant effects of Taurisolo® under different experimental conditions will be included in a future paper. In the present paper, we only wanted to give an indication that Taurisolo® is more effective than other treatments in preventing the formation of free radicals.

Comments 7: Finally, review the English grammar and syntax for the entire text. In fact, there are incorrect English syntax periods as they are too long. Please carefully review the whole text along the article and correct it to make it easier to read. I suggest having the text reread and proofread by a native English speaker.

Response 7: Following the request, the text has been revised by a native English speaker and changes in grammar and syntax have been made at various point in the text. 

Round 2

Reviewer 1 Report

Comments and Suggestions for Authors

The authors well addressed to my concerns except one. The authors should provide the evidence at the molecular level of effects of Taurisolo® in the manuscript.

Author Response

Comments 1: The authors well addressed to my concerns except one. The authors should provide the evidence at the molecular level of effects of Taurisolo® in the manuscript.

Response 1: 

At the moment we have no other molecular evidence of the effects of Taurisolo® other than the biochemical data presented in the previous paper (reference 25), which we have detailed better in the discussion (page: 8; lines: 281 -289).

Any suggestionsare welcome and we will be happy to plan further experiments in the future.

Reviewer 2 Report

Comments and Suggestions for Authors

The revise manuscript has replied my comments item-by-item. I have nor more comments

Author Response

Comments 1:

The revise manuscript has replied my comments item-by-item. I have nor more comments

Response 1:

No request